# New Contributions to the Species Diversity of the Genus *Hydnum* (*Hydnaceae*, *Cantharellales*) in China: Four New Taxa and Newly Recorded Species

**DOI:** 10.3390/jof11060431

**Published:** 2025-06-04

**Authors:** Yong-Lan Tuo, Libo Wang, Xue-Fei Li, Hang Chu, Minghao Liu, Jiajun Hu, Zheng-Xiang Qi, Xiao Li, Yu Li, Bo Zhang

**Affiliations:** 1Engineering Research Center of Edible and Medicinal Fungi Ministry of Education, Jilin Agricultural University, Changchun 130118, China; tuoyonglanfungi@163.com (Y.-L.T.); wlblucky@126.com (L.W.); lixuefei2020@163.com (X.-F.L.); shihanghangzhang@163.com (H.C.); minghaoliu14@163.com (M.L.); qzx7007@126.com (Z.-X.Q.); lxmogu@163.com (X.L.); 2College of Life Sciences, Zhejiang Normal University, Jinhua 321000, China; hujifungi@163.com; 3Sanjiang Fungal Industry Collaborative Innovation Center, Jilin Agricultural University, Changchun 130118, China; 4Industrial Development Institute for Plants, Animals and Fungi Integration of Biyang County, Biyang 463799, China

**Keywords:** *Cantharellales*, *Hydnaceae*, ectomycorrhizal fungi, taxonomy, new taxa

## Abstract

*Hydnum*, a well-defined genus in the family *Hydnaceae* (order *Cantharellales*), is characterized by its distinctive spine-bearing hymenophores. In this study, we performed a multi-locus phylogenetic analysis (ITS-nrLSU-*tef1*) of *Hydnum* species. Integrating morphological examinations and phylogenetic evidence, we identified and delineated five *Hydnum* species in China, which include four novel species (*Hydnum crassipedum*, *H. albomarginatum*, *H. fulvostriatum*, and *H. bifurcatum*) and the first record (*H. orientalbidum*) in Anhui Province. This study provides a comprehensive morphological description (including macroscopic morphology and microscopic structure), hand-drawn illustrations (encompassing basidiocarps, basidiospores, basidia, and pileipellis hyphae), morphological comparative analysis with similar species, and comparative phylogenetic analysis with related taxa. Furthermore, we developed a dichotomous key for identifying *Hydnum* species distributed in China.

## 1. Introduction

The genus *Hydnum* L. (*Hydnaceae* and *Cantharellales*), typified by *H. repandum* L. [1], is microscopically characterized by a monomitic hyphal system, nodose-septate hyphae, 2–6-sterigmate basidia, and subglobose to obovoid-elliptical, smooth, subhyaline, thin-walled basidiospores [2,3]. With transcontinental distribution spanning Asia, Australia, Europe, and North America, species within this genus have been both collected and utilized by humans for centuries [2,4]. Notably, beyond serving as a culinary resource, they establish ectomycorrhizal symbioses with plants in *Fagales* and *Pinaceae*, contributing significantly to forest ecosystem stabilization [2].

Linnaeus established the genus *Hydnum* in 1753, which initially encompassed all fungi with hydnoid hymenophores [5]. Molecular phylogenetic analyses later revealed that the hydnoid hymenophore evolved independently in distantly related lineages [6], leading to the transfer of many former *Hydnum* species to other genera such as *Hericium*, *Phellodon*, and *Sarcodon* [5,6,7]. Early morphology-based studies underestimated *Hydnum* diversity [8,9,10,11] due to extensive intraspecific morphological overlaps, particularly within the *H. rufescens* complex [1,7]. Advancements in molecular systematic studies, however, have progressively revealed greater species richness in this genus: Feng et al. [12] identified 31 *Hydnum* species through multilocus analyses (ITS, RPB1, and TEF1α), delineating four major evolutionary lineages (*Albomagnum*, *Vesterholtii*, *Rufescens*, and *Repandum*). Subsequently, Niskanen et al. [1] elevated the species count to 49 via integrative taxonomy, constructing a hierarchical framework with four subgenera (*Alba*, *Hydnum*, *Pallida*, and *Rufescentia*), four sections (*Hydnum*, *Olympica*, *Magnorufescentia*, and *Rufescentia*), and three subsections (*Mulsicoloria*, *Rufescentia*, and *Tenuiformia*). Building on this, Cao et al. [6] proposed six phylogenetically distinct clades within *Hydnum*: *Rufescentia*, *Hydnum*, *Pallida*, *Brevispina*, *Alba*, and *Alba sensu lato* (s.l.). Collectively, these advances highlight the pivotal role of integrative taxonomy—systematically combining molecular phylogenetics with morphological traits—in resolving taxonomic ambiguities and uncovering hidden biodiversity across *Hydnum* and its related genera.

The genus *Hydnum* exhibits remarkable species diversity, with approximately 60 species recognized worldwide [8,13,14,15,16,17], among which 32 have been described as novel species from Europe and North America. In recent years, research on *Hydnum* diversity in China has progressed rapidly. Prior to 2016, only three species—*H. repandum*, *H. repandum* var. *album*, and *H. rufescens*—had been reported in China [18,19]. In 2016, Feng et al. [12] conducted molecular phylogenetic studies on Chinese *Hydnum* species, which revealed substantial diversity within this genus. Subsequently, Cao et al. [6] performed a phylogenetic investigation of *Cantharellales* in China and described 10 new *Hydnum* species. To date, 29 *Hydnum* species have been confirmed to be distributed in China [6,13,20,21,22]. Previous studies have demonstrated that species within the genus *Hydnum* are predominantly distributed in temperate regions globally, while subtropical areas remain significantly underrepresented in taxonomic records [1,2,7,12,14,23]. This disparity is particularly evident in the Dabie Mountains, a critical transitional zone between subtropical and warm-temperate ecosystems in China [24], which supports exceptional species diversity and constitutes a key biodiversity conservation area [25]. Despite its ecological significance, *Hydnum* species distribution data in this region remain critically deficient, with no comprehensive taxonomic documentation available to date. Continued intensive field investigations in this area are expected to yield new *Hydnum* species discoveries, underscoring the urgency of systematic taxonomic research. This study aims to carry out the following: (i) refine the taxonomic framework of the genus *Hydnum* with particular emphasis on understudied subtropical ecosystems such as the Dabie Mountains and (ii) develop a diagnostic key for Chinese *Hydnum* species to facilitate accurate field identification and inform biodiversity conservation strategies.

## 2. Materials and Methods

### 2.1. Specimen Collecting

Fifteen specimens of *Hydnum* were collected during the rainy season (July to September) in Anhui (114°54′–119°37′ E, 29°41′–34°38′ N) and Jilin Province (121°38′–131°19′ E, 40°50′–46°19′ N), China, from 2019 to 2023. The photographs of the specimens were taken in situ. After thoroughly documenting the fresh macroscopic features, the specimens were placed in a drying oven at 40–45 °C. The dried specimens were deposited in the Mycology Herbarium of Jilin Agricultural University (HMJAU).

### 2.2. Morphological Studies

Micromorphological observations were conducted using a Carl Zeiss Axio Lab A1 compound microscope (Carl Zeiss AG, Jena, Germany). Specimens (collected from each basidiocarp: five spines, approximately 0.3 cm^2^ of pileipellis and 0.3 cm^2^ of stipitipellis) were mounted in 3% (*w*/*v*) KOH solution containing 1% (*w*/*v*) Congo Red for staining. Amyloid reactions were tested with Melzer’s reagent [1.5 g KI, 5 g I_2_, 20 g CCl_3_CH(OH)_2_ (dissolved in 20 mL glycerol)]. Spore dimensions (*n* = 30 per spine) were recorded as (a) b–c (d) (98% values within b–c), with a mean length (avg. L) and width (avg. W) calculated excluding extremes; Q denotes L/W ratio (L: spore length, W: spore width). For ultrastructural analysis [26,27,28], spores were imaged using a JEOL JSM-7900F scanning electron microscope (JEOL Ltd., Tokyo, Japan) at an accelerating voltage of 5.00 kV. Specimens were prepared via silica gel desiccation followed by gold sputter coating (45 s deposition time under a chamber pressure of <1.0 × 10^−^^3^ Pa). Colorimetric parameters were calibrated according to the Methuen Handbook [29].

### 2.3. DNA Extraction, PCR, and Sequencing

The total genomic DNA was extracted using a NuClear Plant Genomic DNA Kit (CW0531M, CoWin Biosciences, Taizhou, China) according to the manufacturer’s instructions. The primer pairs ITS1F/ITS4 [30,31], LROR/LR7 [32], and tef1F/tef1R [12] were employed to amplify and sequence the internal transcribed spacer regions (ITSs), large subunit nuclear ribosomal RNA (nrLSU), and translation elongation factor 1 (*tef1*), respectively. Each PCR reaction (15 μL final volume) contained 1 μL of genomic DNA, 7.5 μL of SanTaq^®^ PCR Master Mix (B532061-0040, Sangon Biotech, Shanghai, China), 4.5 μL of ddH_2_O, and 1 μL each of forward and reverse primer (10 μM). The thermal cycling protocol included an initial denaturation at 95 °C for 3 min, followed by 36 cycles at 95 °C for 40 s, 56 °C for 45 s, and 72 °C for 1.5 min, with a final extension at 72 °C for 8 min (10° C/s ramp rate). PCR products were visualized using UV transillumination after electrophoresis on 1.0% agarose gels stained with ethidium bromide and purified using a Genview High-Efficiency Agarose Gels DNA Purification Kit (Gen-View Scientific Inc., Galveston, TX, USA). Purified products were Sanger sequenced by Sangon Biotech Limited Company (Shanghai, China). Sequences were assembled using Seqman (Lasergene v.7.1, DNASTAR), and the consensus sequences were deposited in GenBank (https://www.ncbi.nlm.nih.gov/genbank/, accessed on 16 March 2025).

### 2.4. Phylogenetic Analyses

A multilocus (ITS-nrLSU-*tef1*) phylogenetic tree was constructed by integrating newly sequenced data and GenBank-derived sequences using PhyloSuite v1.2.3 [33] (Table 1). The workflow included the following: (1) sequence alignment via MAFFT implemented in PhyloSuite; (2) optimal substitution model selection through ModelFinder under the Akaike Information Criterion (HKY + I + G + F for Bayesian Inference; G4 + I + G + F for Maximum Likelihood); (3) Bayesian analysis with 15 million generations (sampling every 100 generations, 25% burn-in, and split frequency < 0.001); (4) Maximum Likelihood analysis with 1000 bootstrap replicates; and (5) topology visualization using the ITOL platform.

## 3. Results

### 3.1. Taxonomy

***Hydnum fulvostriatum*** Y.L.Tuo, B. Zhang & Y. Li sp. nov.; Figure 1a–f, Figure A1 and Figure A2.

Fungal Name: FN 572487.

Etymology: The specific epithet “*fulvostriatum*” is derived from the Latin words *fulvus* (yellowish-brown) and *striatus* (striped), referring to the characteristic yellowish-brown zonate pattern observed on the pileus margin of this species.

Holotype: CHINA. Anhui Province, Lu’an city, Tianma National Nature Reserve, 31°13′16.34″ N, 115°49′59.49″ E, elev. 629.4 m. on soil in *Quercus glauca* Thunb. Forest, 6 August 2023, Yonglan Tuo (FJAU66566). GenBank accession numbers: ITS = PV329849 and LSU = PV356807.

Diagnosis: *H. fulvostriatum* is distinguished from other *Hydnum* species by a yellowish-brown zonate pattern on the pileus margin and globose to subglobose basidiospores (7.0–7.2 × 6.8–7.0 μm).

Description: Basidiocarps medium to large, 26–45 mm in height, solitary to scattered. Pileus 44–65 mm wide, yellowish white to pale orange (4A2–5A3), irregularly round, shallowly depressed in the center, covered with white to yellowish white (4A1–4A2) tomentum, surface dry; margin incurved, thin, pronounced yellow-brown zonate, discoloration not observed. Hymenophore hydnoid, adnate, not decurrent, surface orange–white (4A3); spines conical, brittle, 2–7 mm long, 0.6–1.2 mm diameter, crowded, 2–3 per mm^2^. Stipe central, white (4A1), cylindrical, 30–45 mm long, 8–11 mm wide, solid, surface covered with white (4A1) tomentum. Context fleshy, 5–10 mm thick, dry, discoloration not observed. Odor mild or fruity.

Basidiospores (6.8) 7.0–7.2 (8.0) × (6.5) 6.8–7.0 (7.5) μm, avg. L = 7.2 μm, avg. W = 6.9 μm, Q = 1.00–1.08 (1.17) (n = 60/4), avg. Q = 1.04, globose to subglobose, thin-walled, smooth, hyaline in 3% KOH, some with granular contents or hyaline oily droplets, hilar appendix 0.5–1.0 μm long. Basidia fusiform to suburniform, (35.0) 40.0–46.0 (50.0) × (7.5) 8.0–9.5 (10.5) μm, some with granular contents or hyaline oily droplets; sterigmata 2–4, 4.5–6.0 × 0.5–1.0 µm, conical, thin-walled, smooth. Basidioles numerous, subclavate, smaller than basidia, (20.5) 25.0–30.0 × 7.0–8.0 (9.0) μm, some with granular contents. Cystidia absent. Subhymenium trama filamentous, hyphae 3.5–4.0 μm wide, thin-walled, olive in 3% KOH. Hyphae of spines 5.0–7.0 μm, thin-walled, apex cylindrical. Pileipellis composed of cylindrical hyphae, 6.0–8.5 μm wide, subparallel, occasionally branched; cells 57.0–145.5 × 6.0–8.5 μm, yellowish in 3% KOH, terminal elements dilated at apex. Stipitipellis composed of subcylindrical hyphae, thin-walled hyphae, 6.5–9.0 μm wide. Clamps present in all tissues.

Habitat and distribution: Solitary to gregarious in *Q. glauca* forests; currently known only from the Dabie Mountains, China.

Additional specimens examined: CHINA. Anhui Province, Lu’an city, Tianma National Nature Reserve, 31°14′10.97″ N, 115°50′3.08″ E, elev. 688.5 m. on soil in *Q. glauca* forest, 16 August 2023, Yonglan Tuo (FJAU66567). GenBank accession numbers: ITS = PV329850 and LSU = PV356808.

Notes: Morphologically, *H. fulvostriatum* is closely related to *H. subrufescens*, with medium to large basidiocarps, a pale yellowish-white pileal surface, white to pale cream spines, and a white tomentum stipe. However, it can be distinguished from *H. subrufescens* by its smaller basidiospores (average size: 7.2 × 6.9 vs. 8.1 × 7.0 μm) and its distinctive yellowish-brown striations along the margin of the pileus.

Multigene phylogenetic analysis revealed that the sequences of *H. fulvostriatum* clustered together, forming a distinct lineage. This lineage was identified as a sister clade to *H. roseotangerinum*, but it exhibits a more distant phylogenetic relationship with *H. subrufescens*. Based on the morphological characteristics provided above and the phylogenetic results, *H. fulvostriatum* should be classified as a member of the subsect. *Rufescentia*.

***Hydnum crassipedum*** Y.L.Tuo, B. Zhang & Y. Li sp. nov. Figure 2a–f, Figure A1 and Figure A2.

Fungal Name: FN 572486.

Etymology: The species epithet “*crassipedum*” is derived from Latin *crassus* (thick) and *pes* (foot), referring to the characteristically robust stipe of this species.

Holotype: CHINA. Anhui Province, Lu’an city, Tianma National Nature Reserve, 31°20′1.13″ N, 115°54′6.53″ E, elev. 636.1 m. on the ground of *Quercus variabilis* Blume forest, 28 September 2023, Yonglan Tuo (FJAU66572). GenBank accession numbers: ITS = PV329853, *tef1* = PP357260, and LSU = PV356811.

Diagnosis: *H. crassipedum* can be distinguished from other *Hydnum* species by its thicker stipe (15–25 mm), brownish-yellow tomentum pileus surface, and subglobose to broadly ellipsoid basidiospores measuring 8.0–8.5 × 7.0–7.5 µm.

Description: Basidiocarps medium, 32–45 mm in height, fleshy, and usually gregarious. Pileus 24–56 mm wide, yellowish-white to pale orange (4A2–5A3) when fresh, irregularly round, plano-convex when young, center shallowly depressed in age, discoloration not observed. Margin incurved, covered with white to pale orange (4A1–5A3) tomentum when young. Context 6–12 mm thick, white (4A1), odor mild or slightly sweet, discoloration not observed. Hymenophore spinose, decurrent, surface white to yellowish white (4A1–4A2), shorter near the pileus margin, cylindrical, slightly pointed tip, 2–5 mm long, 0.5–1 mm diameter, sparse, 1–2 per mm^2^. Stipe central or eccentric, white (4A1), 21–45 mm long, 15–25 mm wide, subcylindrical, slightly enlarged downwards, solid, covered with white (4A1) tomentum, discoloration not observed. Odor mild or fruity.

Basidiospores (7.0) 8.0–8.5 (9.0) × (6.5) 7.0–7.5 (8.0) µm (n = 60/2), avg. L = 8.0 µm, avg. W = 6.99 µm, Q = 1.11–1.17 (1.23), avg. Q = 1.15, subglobose to broadly ellipsoid, smooth, thin-walled, hyaline in 5% KOH, with finely granulose contents, hilar appendix 0.5–1 μm long. Basidia subcylindric or subclavate, (36.0) 37.5–45.5 × 9.0–9.5 (11.0) µm, sometimes with finely granulose contents; sterigmata 2–4, up to 4.5–5.5 μm long. Basidioles numerous, subclavate, smaller than basidia, 32.0–35.0 × 8.0–10.0 μm. Cystidia absent. Subhymenium trama filamentous, 3.0–4.0 μm wide, cylindrical, thin-walled, subparallel, pale yellow in 3% KOH. Hyphae of spines 5.0–7.0 µm wide. Pileipellis composed of cylindrical hyphae, 6.0–9.0 μm wide, thin walled, subparallel, rarely branched; cells, 67.0–125.0 × 6.0–9.0 μm, terminal elements rounded at apex. Stipitipellis composed of cylindrical hyphae, thin walled, 4.0–4.5 μm wide, terminal elements rounded at apex. Clamps present in all tissues.

Additional specimens examined: CHINA, Anhui Province, Lu’an city, Tianma National Nature Reserve, 31°20′4.38″ N, 115°54′6.69″ E, elev. 627.5 m, on the ground of *Q. variabilis* forest, 15 September 2023, Yonglan Tuo (FJAU66573). GenBank accession numbers: ITS = PV329854, *tef1* = PP357261, and LSU = PV356812.

Habitat and distribution: Growing gregariously in *Q. variabilis* forest. Currently known only from the Dabie Mountains, China.

Notes: Morphologically, *H. crassipedum* is similar to *H. erectum*. However, the stipe of *H. crassipedum* is thicker (15–25 vs. 1.2–1.5 mm), and its basidiospores are larger (average 8.0 × 6.99 vs. 7.67 × 7.09 µm).

Phylogenetic analysis revealed that *H. crassipedum* forms a distinct lineage and is closely related to *H. roseotangerinum*. Based on the morphological characteristics described above and the phylogenetic results, *H. crassipedum* should be classified as a member of subg. *Rufescentia*.

***Hydnum albomarginatum*** Y.L.Tuo, B. Zhang & Y. Li sp. nov.; Figure 3a–f, Figure A1 and Figure A2.

Fungal Name: FN 572488.

Etymology: The specific epithet “*albomarginatum*” is derived from Latin *albus* (white) and *marginatus* (edged), referring to the distinctive white marginal zone characteristic of this species.

Holotype: CHINA. Anhui Province, Lu’an city, Tianma National Nature Reserve, 31°18′58.24″ N, 115°55′32.23″ E, elev. 751.2 m. on the ground of *Quercus serrata* Thunb. forest, 22 October 2023, Yonglan Tuo (FJAU66574). GenBank accession numbers: ITS = PV329855, tef1 = PP357262, and LSU = PV356813.

Diagnosis: *H. albomarginatum* differs from other *Hydnum* species by having a white tomentum along the margin of the pileus, forming characteristic white striations, and globose basidiospores measuring 8.0–8.5 × 7.5–8.0 μm.

Description: Basidiocarps small to medium, fleshy, 30–45 mm in height, usually gregarious. Pileus 23–66 mm wide, covered with yellowish white (4A2) tomentum, irregularly round, shallowly depressed in the center; margin incurved, and covered white (4A1) tomentum, forming characteristic white zonate, discoloration not observed. Hymenophore hydnoid, spines not decurrent, conic, orange–white (4A3), 2–6 mm long, 0.5–1.0 mm diameter, sparse, 1–2 per mm^2^. Stipe 26–50× 8–17 mm, central, cylindrical, hollow, slightly enlarged at the base, white to orange–white (4A1–4A3), surface covered white (4A1) tomentum. Context 5–10 mm thick, dry, fleshy. Odor mild or fruity. Discoloration not observed.

Basidiospores (7.0) 8.0–8.5 (9.0) × (7.0) 7.5–8.0 (8.5) μm (n = 60/2), avg. L =8.07 μm, avg. W = 7.92 μm, Q = 1.00–1.09 (1.18), avg. Q = 1.01, globose, thin-walled, smooth, hyaline in 3% KOH, some with granular contents or hyaline oily droplets; hilar appendix 0.2–0.5 μm long. Basidia clavate to suburniform, (30.5) 33.5–40.0 (42.5) × (9.0) 10.0–10.5 (11.0) μm, some with granular contents or hyaline oily droplets; sterigmata 2–4, 5.0–6.5 × 1.5–2.0 µm, conical, thin-walled, and smooth. Basidioles numerous, subclavate, smaller than basidia, (22.0) 25.0–30.0 (31.0) × (7.0) 8.0–9.0 (11.0) μm, some with granular contents. Cystidia absent. Subhymenium trama filamentous, hyphae 4.0–6.0 μm wide, thin-walled, olive in 3% KOH. Hyphae of spines 6.5–8.0 μm, thin-walled, apex cylindrical. Pileipellis, subparallel to slightly interwoven; cells 40.0–160.0 × 5.0–8.0 μm, terminal elements conical at apex. Stipitipellis composed of cylindrical hyphae, slightly interwoven, 6.0–7.0 μm wide, terminal elements rounded at apex. Clamp connections present.

Habitat and distribution: Growing solitarily or gregariously in *Q. serrata* forest. Currently known only from the Dabie Mountains, China.

Additional specimens examined: CHINA, Anhui Province, Lu’an city, Tianma National Nature Reserve, 31°18′50.13″ N, 115°55′32.96″ E, elev. 764.5 m, 11 October 2023 on the ground of *Q. serrata* forest, Yonglan Tuo (FJAU66575). GenBank accession numbers: ITS = PV329856, tef1 = PP357263, and LSU = PV356814.

Notes: Morphologically, although *H. albomarginatum* and *H. berkeleyanum* both exhibit smaller pileus diameters and globose basidiospores, they can be distinguished by the smaller basidiospore size of the former (8.07 × 7.92 vs. 8.5 × 7.95 µm). Additionally, *H. albomarginatum* features a yellowish pileus with distinct white striations along the margin, whereas *H. berkeleyanum* typically displays a pale orange to ochraceous-brown pileus lacking zonate patterns.

Phylogenetic analysis showed that *H. albomarginatum* forms a distinct lineage. This lineage exhibits a more distant phylogenetic divergence with *H. berkeleyanum*. Based on the morphological characteristics given above and the phylogenetic results, *H. albomarginatum* should be classified as a member of the subsect. *Rufescentia.*

***Hydnum bifurcatum*** Y.L.Tuo, B. Zhang & Y. Li sp. nov.; Figure 4a–f, Figure A1 and Figure A2.

Fungal Name: FN572475.

Etymology: The specific epithet “*bifurcatum*” derives from Latin *bi-* (two) and *furca* (fork), referring to the species’ characteristic bifurcated hymenial spines.

Holotype: CHINA. Jilin Province, Ji’an city, Wunvfeng National Forest Park, 41°16′40″ N, 126°7′5″ E, elev. 810 m, on the ground of *Quercus mongolica* Fischer ex Ledebour forest, 12 September 2019, Yonglan Tuo (FJAU66562). GenBank accession numbers: ITS = PV329845 and *tef1* = PP357252.

Diagnosis: *H. bifurcatum* differs from other *Hydnum* species by its larger pileus (65–135 mm), larger basidiospores (8.5–9.5 × 8–9.5 µm), and bifurcated spines.

Description: Basidiocarps medium to large, 70–110 mm in height, solitary or gregarious. Pileus 65–135 mm wide, orange–white to pale orange (4A3–5A3), irregularly round, surface convex to depressed, discoloration not observed; margin incurved, covered with white (4A1) tomentum when young. Context 6–15 mm thick, white (4A1), brittle, odor mild or slightly sweet, discoloration not observed. Hymenophore spinose, adnate, decurrent; spines, subulate, orange–white (4A3), shorter near the pileus margin, 2–7 mm long, 0.6–1.2 mm diameter, sparse, 1–2 per mm^2^. Stipe central, 30–50 mm long, 15–40 mm wide, subcylindrical, slightly enlarged downwards, solid; surface covered white (4A1) tomentum.

Basidiospores (8.0) 8.5–9.5 (10.0) × (7.5) 8.0–9.5 (10.0) µm, avg. L = 9.12 µm, avg. W = 8.97 µm, Q = 1.00–1.06 (1.11), avg. Q = 1.01, globose, smooth, thin-walled, hyaline, some with granular contents; hilar appendix 0.5–1.0 μm long. Basidia 50.0–60.0 × 10.0–11.0 µm, clavate, sometimes with finely granulose contents; sterigmata up to 4.0–5.0 µm long, 2–4 sterigmata. Basidioles numerous, subcylindrical or subclavate, smaller than basidia, 40.0–52.0 × 8.0–10.0 μm. Cystidia absent. Subhymenium trama filamentous, 3.0–4.0 μm wide, subcylindrical, thin-walled, subparallel, rarely branched hyphae, extend along the surface of the spines. Hyphae of spines 6.0–10.0 µm wide, thin-walled, hyaline in 5% KOH. Pileipellis composed of cylindrical hyphae, 8–10 μm wide, thin-walled, densely interwoven to subparallel, rarely branched; cells, 80.0–170.0 × 8.0–11.5 μm, terminal slightly dilated at apex. Stipitipellis composed of subcylindrical hyphae, thin-walled, 4.0–7.0 μm wide. Clamp connections present.

Additional specimens examined: CHINA. Jilin Province, Ji’an city, Wunvfeng National Forest Park, 41°16′33″ N, 126°7′3″ E, elev. 856 m, on the ground of *Q. mongolica* forest, 16 September 2020, Yonglan Tuo (FJAU66563). GenBank accession numbers: ITS = PV329846 and *tef1* = PP357253.

Habitat and distribution: Occurs solitarily in *Q. mongolica* forest; currently known only from Jilin Province, China.

Notes: Morphologically, *H. bifurcatum* can easily be misidentified as *H. tomaense*. However, *H. bifurcatum* is distinguished by its bifurcated spines, larger basidiocarps (65–135 vs. 40–100 mm), and larger basidiospores (avg. = 9.12 × 8.97 vs. 7.5–8.5 × 7–8 µm). Notably, bifurcated spines are rare among *Hydnum* species.

The multilocus phylogenetic analyses revealed that the two Chinese *H. bifurcatum* specimens form a monophyletic clade, exhibiting close phylogenetic relationships with *H. tomaense*, *H. treui*, and *H. zongolicense*. Based on the morphological characteristics described earlier and these phylogenetic findings, we propose that *H. bifurcatum* should be classified within subg. *Alba s.l*.

***Hydnum orientalbidum*** R. Sugaw. & N. Endo.; Figure 5a–f, Figure A1 and Figure A2.

Basidiocarps small to medium sized, solitary to scattered. Pileus 30–45 mm wide, convex to subapplanate, smooth, azonate, white to yellowish white (4A1–4A2), and covered white (4A1) tomentum. Context 2–3 mm thick, white (4A1), discoloration not observed when exposed. Hymenophore hydnoid, spines fleshy, non-decurrent, subulate, surface white (4A1), 1–3 mm long, 0.25–0.5 mm diameter, crowded, 2–4 pre mm^2^. Stipe central to slightly eccentric, white (4A1), 20–35 mm long, 8–12 mm wide, subcylindrical, solid, basal mycelium white.

Basidiospores (4.0) 4.5–5 (6.0) × (4.0) 4.5–5 (6.0) μm, avg. L = 4.6 μm, avg. W = 4.4 μm, Q = 1.0–1.11 (n = 60/2), avg. Q = 1.05, globose, thin-walled, smooth, hyaline in 3% KOH, some with granular contents; hilar appendix 0.5 μm long. Basidia clavate to suburniform, (30.0) 32.5–38.5 (44.5)× (6.0) 6.5–7.5 (9.0) μm, some with granular contents and hyaline oily droplets, sterigmata 2–6, 3.5–4.0 × 2.0–6.5 µm, conical, thin-walled, smooth. Basidioles numerous, smaller than basidia, (20.0) 25.0–30.0 (42.0) × (5.0) 6.0–7.0 (9.0) μm, some with granular contents. Cystidia absent. Subhymenium trama filamentous, hyphae 6.5–7.0 μm wide, thin-walled, hyaline in 3% KOH. Hyphae of spines 3–4 μm, thin-walled, apex cylindrical. Pileipellis composed of cylindrical hyphae, subparallel, terminal elements cylindrical at apex, 5.5–8.0 μm. Stipitipellis composed of cylindrical hyphae, slightly interwoven, 3.0–5.5 μm wide, terminal elements rounded at apex. Clamp connections present.

Habitat and distribution: The species grows either solitarily or gregariously in *Q. serrata* forest. Currently documented in Japan, the Province of Anhui, Sichuan, and Zhejiang provinces in China.

Specimens examined: China, Anhui Province, Lu’an city, Tianma National Nature Reserve, 31°15′39.37″ N, 115°41′47.35″ E, elev. 1129.6 m, 5 October 2023, Yonglan Tuo (FJAU66570, ITS = PV329857, tef1 = PP357258, LSU = PV356809). Anhui Province, Lu’an city, Tianma National Nature Reserve, 31°15′37.09″ N, 115°41′54.55″ E, elev. 1055.1 m, 15 October 2023, Yonglan Tuo (FJAU66571, ITS = PV329858, tef1 = PP357259, LSU = PV356810).

Notes: The Anhui specimens share consistent characteristics with previous descriptions of *H. orientalbidum* [7,22]; however, the habitat and distribution differ from those in the present study (*Q. serrata* vs. *Picea glehnii* forest).

Phylogenetic analyses based on multilocus datasets indicate that two specimens from Anhui are well nested within the *H. orientalbidum* clade.

### 3.2. Molecular Phylogeny

Phylogenetic analysis based on the concatenated ITS-nrLSU-*tef1* dataset (comprising 170 sequences) revealed congruent topologies between Maximum Likelihood (ML) and Bayesian Inference (BI) methods (BI tree only). The resulting phylogeny was divided into six major clades: Subgenus *Rufescentia*, subgenus *Hydnum*, subgenus *Pallida*, subgenus *Brevispina*, subgenus *Alba*, and subgenus *Alba s.l.* (Figure 6 and Figure 7).

Within subg. *Rufescentia*, two sections were identified: sect. *Magnorufescentia* and sect. *Rufescentia*, the latter being further subdivided into four subsections (subsect. *Tenuiformia*, subsect. *Mulsicoloria*, subsect. *Rufescentia*, and subsect. *Ovoideispora*). Subg. *Hydnum* contained sect. *Hydnum* and sect. *Olympica*.

The newly generated sequences resolved five independent, well-supported species-level clades (PP = 0.95–1.0; BS = 98–100%). Notably, three distinct lineages (FJAU66566/FJAU66567; FJAU66572/FJAU66573; FJAU66574/FJAU66575) were resolved as subsect. *Rufescentia* (subg. *Rufescentia*). Two distinct lineages (FJAU66562/FJAU66563; FJAU66570/FJAU66571) were resolved as subg. *Alba s.l.*

## 4. Discussion

Through integrated morphological and molecular analyses, this study confirmed and characterized five *Hydnum* species in China, including four novel taxa (*H. bifurcatum*, *H. crassipedum*, *H. albomarginatum*, and *H. fulvostriatum*) and a newly recorded species (*H. orientalbidum*) from Anhui Province. Currently, 33 *Hydnum* species have been documented in China, 22 of which are only known so far in this country [6,12,13,21,22]. Table A1 (see Appendix B) provides a comparative summary of the key morphological characteristics and ecological information for the *Hydnum* species identified in China.

In our study, three *Hydnum* species—*H. crassipedum, H. albomarginatum,* and *H. fulvostriatum*—are characterized by small to medium-sized basidiomata that are ochraceous to orange-brown in color, with Q values ranging from 1.00 to 1.14. According to the taxonomic key established by Niskanen et al. [1] for the genus *Hydnum*, these species should be classified under the subsection *Magnorufescentia* (Q = 1.07–1.13). However, by integrating the latest phylogenetic findings from Niskanen et al. [1] and Cao et al. [6], along with newly generated sequences from this study, we demonstrate that *H. crassipedum*, *H. albomarginatum*, and *H. fulvostriatum* form well-supported monophyletic clades within the subsection *Rufescentia*. This incongruence between the morphological taxonomy and molecular phylogeny necessitates a re-evaluation of the diagnostic criteria for these subsections. Based on phylogenetic analysis and the morphological characterization of the subsection *Rufescentia* (including basidiocarp morphology, basidiospore shape, and size parameters), we propose that the Q-value boundary for basidiospores in the subsection *Rufescentia* should be expanded beyond the threshold (Q = 1.15–1.30) previously defined by Niskanen et al. [1].

Both *H. bifurcatum* and *H. orientalbidum* are classified within subg. *Alba s.l.*, consistent with previous taxonomic studies [1,6]. However, they exhibit distinct morphological differences: *H. bifurcatum* possesses a comparatively larger, yellowish pileus (diameter 65–135 mm), whereas *H. orientalbidum* is characterized by a whitish pileus. Notably, the basidiospores of *H. bifurcatum* are approximately twice the size of those in *H. orientalbidum* (9.12 × 8.97 µm vs. 4.6 × 4.4 µm). Phylogenetic analyses further reveal that these two species form independent clades, indicating substantial genetic divergence. This evidence suggests that their current classification under the same subgenus (*Alba s.l.*) may require revision, potentially meriting their recognition as distinct subgenera. However, such taxonomic adjustments would require validation through comprehensive morphological comparisons.

Spore characteristics (shape, size, average length, average width, and Q value) are reliable indicators for interspecific identification among species within the genus *Hydnum* [8,9,10,11]. Based on these features, the new taxa described in this study can be clearly distinguished from similar species: *H. bifurcatum* vs. *H. tomaense* (average L × W = 9.12 × 8.97 vs. 7.5–8.5 × 7.0–8.0 µm); *H. crassipedum* vs. *H. erectum* (average L × W = 8.0 × 6.99 vs. 7.67 × 7.09 µm); *H. fulvostriatum* vs. *H. subrufescens* (average L × W = 7.2 × 6.9 vs. 8.1 × 7.0 µm), and *H. albomarginatum* vs. *H. berkeleyanum* (average L × W = 8.07 × 7.92 vs. 8.4 × 7.95 µm). Furthermore, the new taxa can be readily differentiated from their congeners when assessed alongside macro-morphological characteristics, including pileus size, coloration, surface texture, spine dimensions, and spine morphology [1]. For instance, *H. bifurcatum* can be readily distinguished from species exhibiting bifurcated spine structures through an analysis of spore morphology and dimensions. Similarly, *H. fulvostriatum* and *H. albomarginatum* exhibit distinctive brown and white tomentum annulations at the pileus margin, respectively, which serve as diagnostic characters that separate them from other species in the genus. In addition, under the scanning electron microscope (SEM), we observed that the spores of five *Hydnum* species exhibited nearly smooth surfaces with distinct fish scale-like ornamentations of varying depths (Appendix A: Figure A1). This feature has often been overlooked in previous studies, as nearly all prior research findings indicated that spore surfaces in the genus *Hydnum* are smooth [1,2,6,7,9,13,15,16,17,21,22,41]. This discrepancy is understandable, as the resolution limitations of optical microscopes may affect observation accuracy. Therefore, our findings suggest that spore ornamentation patterns could represent a significant taxonomic characteristic for species identification within the genus *Hydnum*. However, further validation is required to confirm their diagnostic utility.

The distribution of most *Hydnum* species within their host flora appears to be limited [12]. In the Northern Hemisphere, particularly in Europe and North America, these species primarily associate with plant hosts in the *Pinaceae* and *Fagaceae* families [23,42]. Our newly identified species are distributed in *Quercus* forest and typically emerge in early autumn in Northeast China and Anhui Province, suggesting a potential linkage to *Quercus* presence and seasonal factors [43,44]. Although 28 *Hydnum* species have been recorded in China, they are rarely reported in *Quercus*-dominated forests [1,6,45,46]. This scarcity may stem from two interrelated factors: on the one hand, most *Quercus* species have restricted distributions, especially those forming homogeneous forest communities [47,48,49]; on the other hand, recorded specimens primarily appeared 3–5 days post rainfall during autumn, likely representing their optimal growth phase [50,51,52]. Beyond this window, dry weather and soil moisture evaporation may suppress spore germination and mycelial growth [53], thereby reducing detection opportunities for ideal habitats and temporal windows.

## 5. Conclusions

Based on morphological observations (basidiocarps size, pileus color, basidiospores dimensions, etc.), *H*. crassipedum, *H. albomarginatum*, and *H. fulvostriatum* can be effectively assigned to sect. *Rufescentia*, whereas *H. orientalbidum* and *H. bifurcatum* are classified into subgenus *Alba s.l*. These taxonomic conclusions were consistently validated through molecular phylogenetic analyses. Comparative morphological analyses (basidiospores morphology and dimensions, spines characteristics, pileus zonate, etc.) further demonstrated that the five *Hydnum* species in this study could be clearly distinguished from related taxa within the genus *Hydnum*. Phylogenetic trees inferred from ITS, nrLSU, and TEF1α sequence data provided robust support for these classifications (BS = 98–100, PP = 0.98–1.0). However, inconsistencies arise in subsect. assignments: phylogenetically, *H. crassipedum*, *H. albomarginatum*, and *H. fulvostriatum* cluster within subsect. *Rufescentia*, whereas morphologically, their Q values (1.00–1.14) fall below the diagnostic threshold for the subsect. *Rufescentia* (1.15–1.30), but align closely with those of subsect. *Magnorufescentia* (Q = 1.07–1.13). To resolve this discrepancy, we propose revising the Q value criterion for subsect. *Rufescentia*. These findings increase the total number of *Hydnum* species in China to 33, 22 of which are only known so far in this country. This study expands the known species distribution in temperate regions, fills a taxonomic gap in the Dabie Mountains, and refines the classification system of *Hydnum*.

### Key to Species of Hydnum in China

1.Basidiomata more or less white to cream yellow21.Basidiomata yellow to orange112.Pileus white32.Pileus cream yellow73.Pileus < 30 mm wide
*H. flavidocanum*
3.Pileus > 30 mm wide44.Habitat in broad-leaved forests54.Habitat in Fagaceous forests65.Pileus > 60 mm wide
*H. cremeoalbum*
5.Pileus < 60 mm wide
*H. orientalbidum*
6.Basidiospores < 5 μm long on average
*H. minus*
6.Basidiospores > 5 μm long on average
*H. treui*
7.Pileus > 60 mm wide87.Pileus < 60 mm wide98.Habitat in broad-leaved forests
*H. roseoalbum*
8.Habitat in *Quercus mongolica* forests
*H. bifurcatum*
9. Spines < 3 mm long109. Spines > 3 mm long
*H. albomagnum*
10.Spines pale orange
*H. pinicola*
10.Spines cream yellow
*H. minum*
11.Basidiomata yellowish-white1211.Basidiomata orange1712.Basidiospores < 6 μm long on average1312.Basidiospores > 6 μm long on average1413.Habitat in angiosperm forests
*H. brevispinum*
13.Habitat in mixed forests
*H. microcarpum*
14.Basidiospores < 7.5 μm long on average1514.Basidiospores > 7.5 μm long on average1615.Spines 1–3 mm long
*H. tenuistipitum*
15.Spines 2–7 mm long
*H. fulvostriatum*
16.Habitat in *Q*. *variabilis* forests
*H. crassipedum*
16.Habitat in *Q*. *serrata* forests
*H. albomarginatum*
17.Basidiospores 7–8 μm long on average1817.Basidiospores > 8 μm long on average1918.Habitat in coniferous forests
*H. jussii*
18.Habitat in Fagaceous forests
*H. erectum*
19.Basidiospores 8–9 μm long on average2019.Basidiospores > 9 μm long on average2720.Spines white2120.Spines orange–white or cream-yellow2221.Spines < 3 mm long
*H. sphaericum*
21.Spines > 3 mm long
*H. sinorepandum*
22.Spines cream-yellow
*H. cremeum*
22.Spines orange–white2323.Basidiospores 8.1–8.5 μm long on average2423. Basidiospores > 8.5 μm long on average2624. Spines < 2 mm long
*H. vesterholtii*
24.Spines > 2 mm long2525.Pileus < 50 mm wide
*H. tangerinum*
25.Pileus > 50 mm wide
*H. berkeleyanum*
26.Basidiospores Q < 1.2
*H. ventricosum*
26.Basidiospores Q > 1.2
*H. pallidomarginatum*
27.Pileus < 50 mm wide2827.Pileus > 50 mm wide
*H. roseotangerinum*
28.Pileus orange–white2928.Pileus light yellow3129.Habitat in mixed forests
*H. flabellatum*
29.Habitat in broad forests3030.Basidiospores > 9.5 μm long on average
*H. longibasidium*
30.Basidiospores < 9.5 μm long on average
*H. longipes*
31.Basidiospores Q > 1.3
*H. melitosarxm*
31.Basidiospores Q < 1.33232.Habitat in *Pinus* forests
*H. pallidocroceum*
32.Habitat in mixed forests
*H. flavoquamosum.*


## Figures and Tables

**Figure 1 jof-11-00431-f001:**
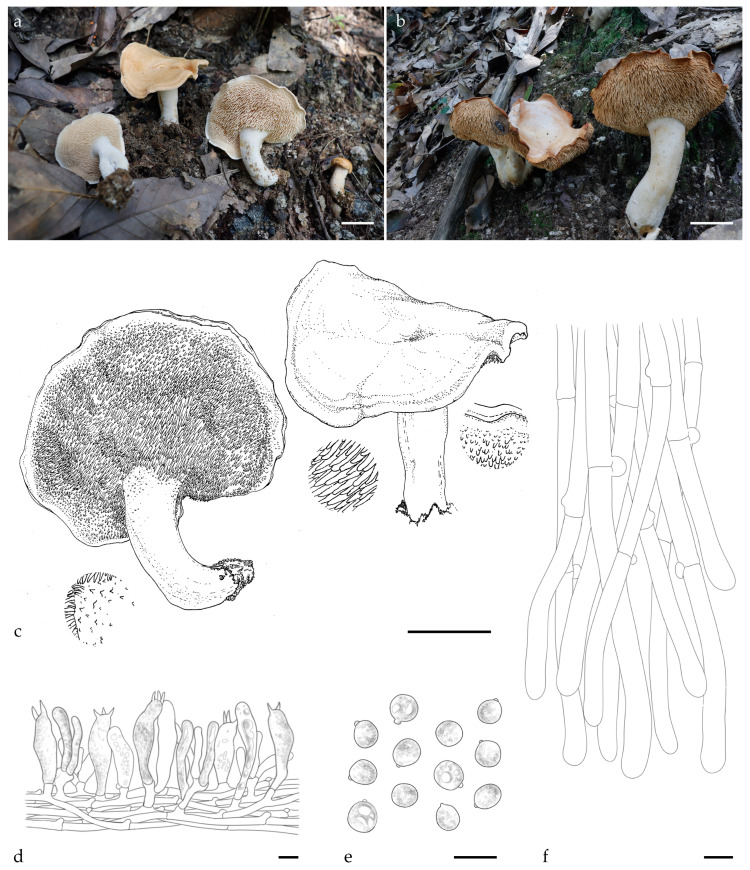
Macroscopic and microscopic features of *H. fulvostriatum* (FJAU66566, holotype). (**a**–**c**) Basidiomata; (**d**) hymenium and subhymenium; (**e**) basidiospores; and (**f**) pileipellis. Scale bar: (**a**–**c**) = 2 cm; (**d**–**f**) = 10 μm.

**Figure 2 jof-11-00431-f002:**
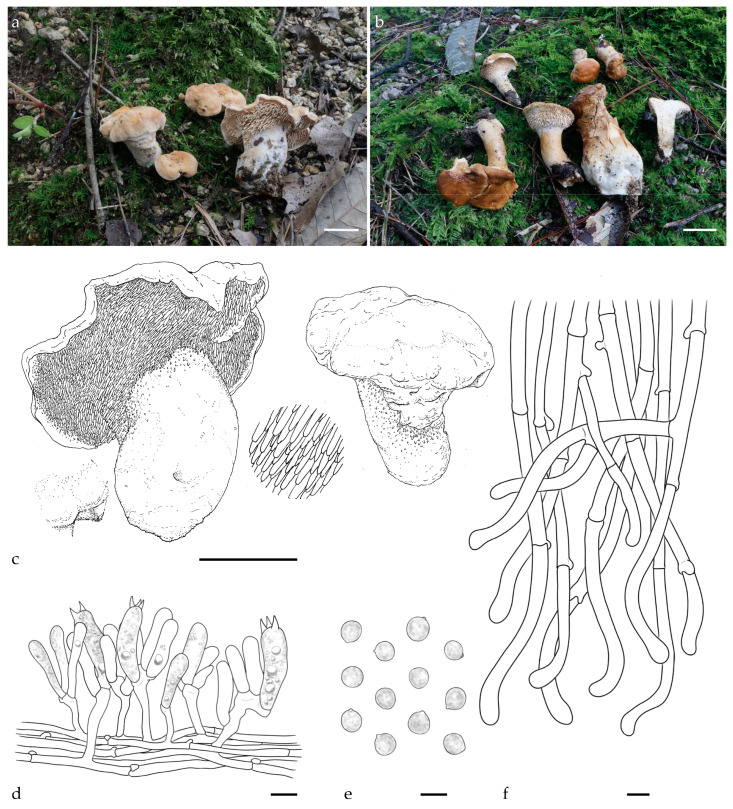
Macroscopic and microscopic features of *H. crassipedum* (FJAU66572, holotype). (**a**–**c**) Basidiomata; (**d**) hymenium and subhymenium; (**e**) basidiospores; (**f**) and pileipellis. Scale bar: (**a**–**c**) = 2 cm; (**d**–**f**) = 10 μm.

**Figure 3 jof-11-00431-f003:**
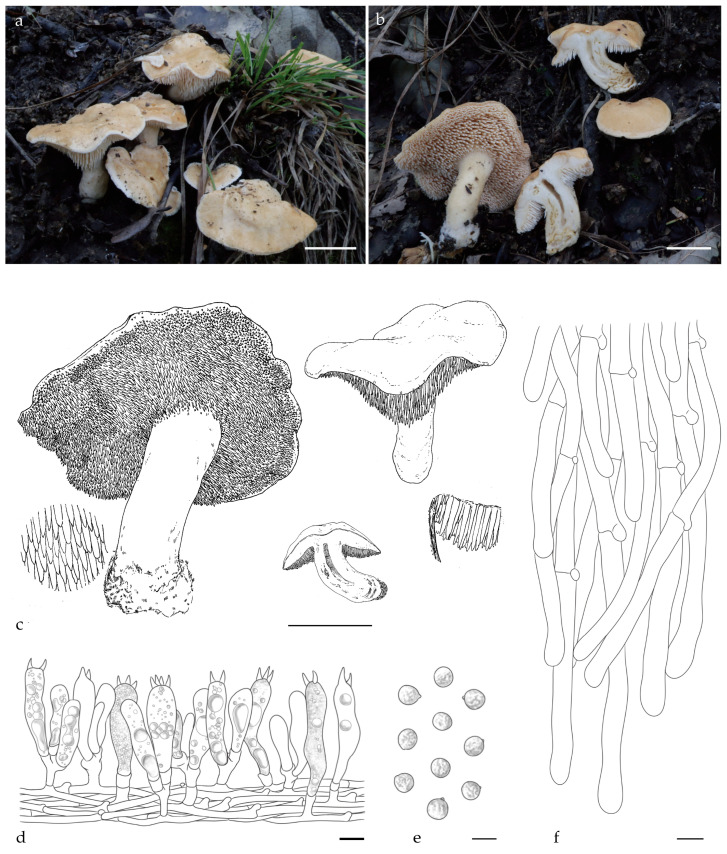
Macroscopic and microscopic features of *H. albomarginatum* (FJAU66574, holotype). (**a**–**c**) Basidiomata; (**d**) hymenium and subhymenium; (**e**) basidiospores; (**f**) and pileipellis. Scale bar: (**a**–**c**) = 2 cm; (**d**–**f**) = 10 μm.

**Figure 4 jof-11-00431-f004:**
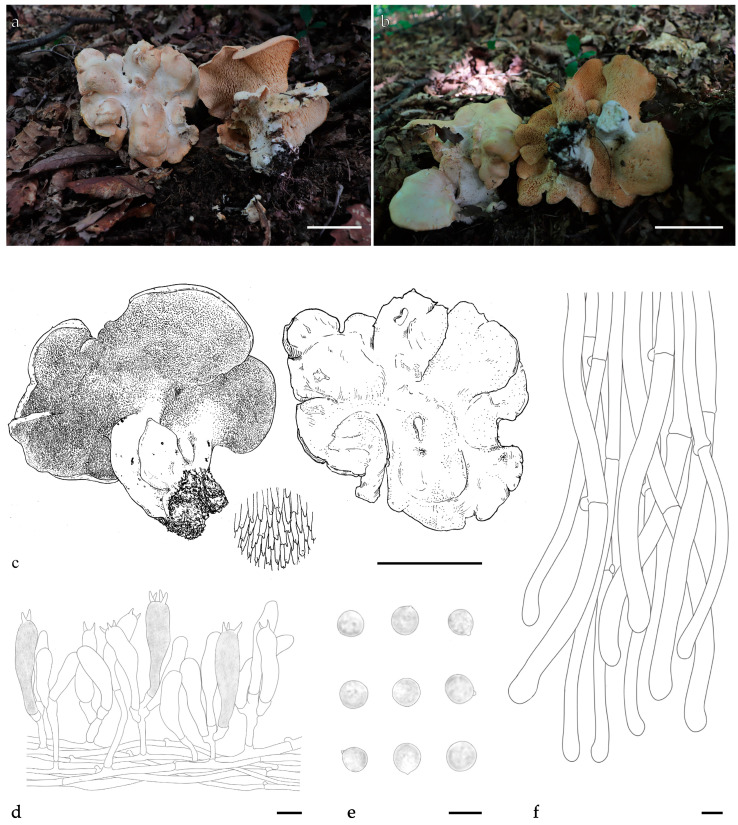
Macroscopic and microscopic features of *H. bifurcatum* (FJAU66562, holotype). (**a**–**c**) Basidiomata; (**d**) hymenium and subhymenium; (**e**)basidiospores; (**f**) and pileipellis. Scale bar: (**a**–**c**) = 5cm; (**d**–**f**) = 10 μm.

**Figure 5 jof-11-00431-f005:**
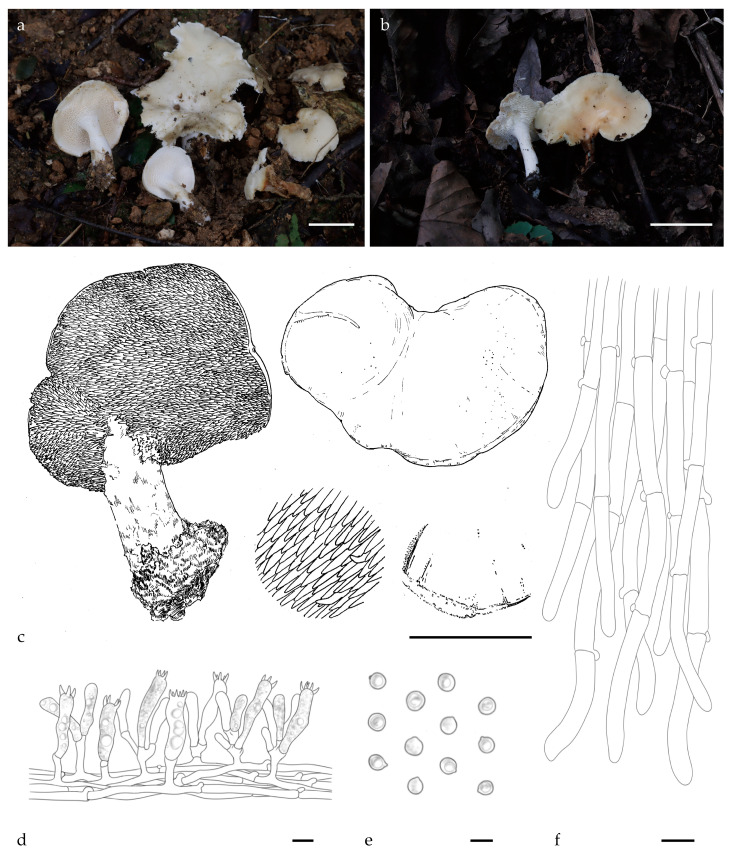
Macroscopic and microscopic features of *H. orientalbidum* (FJAU66574). (**a**–**c**) Basidiomata; (**d**) hymenium and subhymenium; (**e**) basidiospores; and (**f**) pileipellis. Scale bar: (**a**–**c**) = 2 cm; (**d**–**f**) = 10 μm.

**Figure 6 jof-11-00431-f006:**
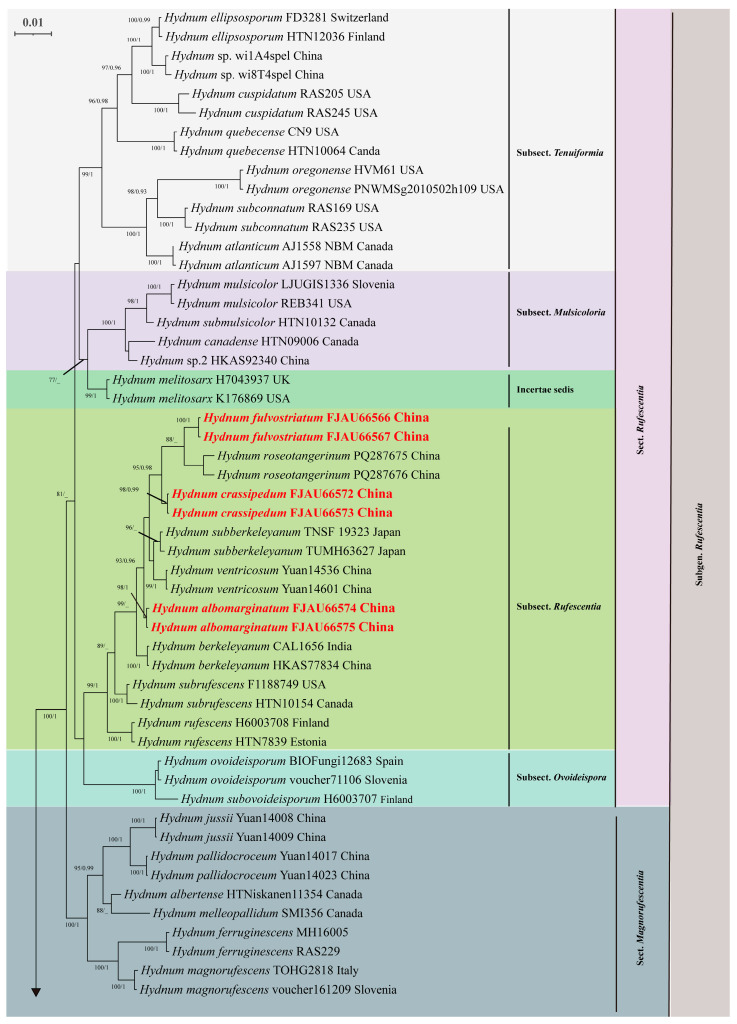
Phylogenetic tree of *Hydnum* inferred through Bayesian Inference and Maximum Likelihood analyses based on the combined nrLSU, ITS, and *tef1* dataset. Node support is indicated as bootstrap (BS) > 70% and posterior probability (PP) > 0.95. Sequences generated in this study are in bold; new taxa are highlighted in red; and new records are indicated in black.

**Figure 7 jof-11-00431-f007:**
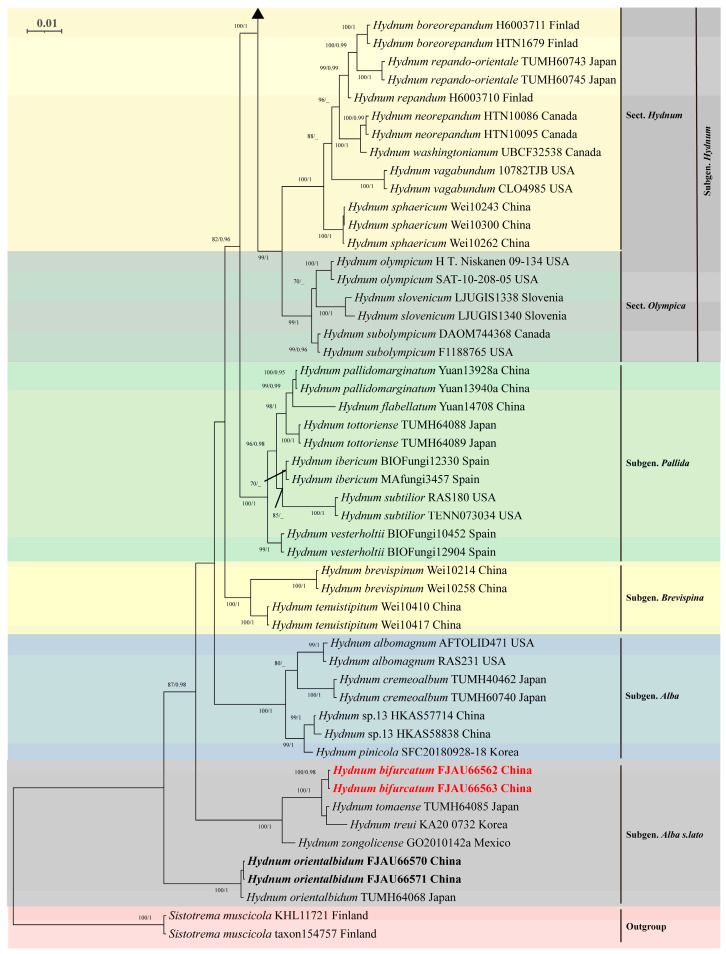
Phylogenetic tree of *Hydnum* inferred through Bayesian Inference and Maximum Likelihood analyses based on the combined *nr*LSU, ITS, and *tef1* dataset. Node support is indicated as bootstrap (BS) > 70% and posterior probability (PP) > 0.95. Sequences generated in this study are in bold; new taxa are highlighted in red; and new records are indicated in black.

**Table 1 jof-11-00431-t001:** Specimens and sequences used in this study.

Species	Specimen Voucher	GenBank No.			Country	References
		nrLSU	ITS	*tef1*		
*H. albertense*	H T. Niskanen 11-354	-	KX388664	-	Canada	[1]
*H. albomagnum*	AFTOL-ID 471	AY700199	DQ218305	DQ234568	USA	[2]
*H. albomagnum*	RAS231	-	MH379943	-	USA	[2]
** *H. albomarginatum* **	**FJAU66574**	**PV356813**	**PV329855**	**PP357262**	**China**	**This study**
** *H. albomarginatum* **	**FJAU66575**	**PV356814**	**PV329856**	**PP357263**	**China**	**This study**
*H. atlanticum*	AJ1597	-	OQ235218	OQ236553	Canada	[34]
*H. atlanticum*	AJ1558	-	OQ235214	OQ236551	Canada	[34]
*H. berkeleyanum*	CAL 1656	NG070500	NR158533	-	India	[35]
*H. berkeleyanum*	HKAS77834	KU612667	KU612525	-	China	[12]
** *H. bifurcatum* **	**FJAU66562**	**-**	**PV329845**	**PP357252**	**China**	**This study**
** *H. bifurcatum* **	**FJAU66563**	**-**	**PV329846**	**PP357253**	**China**	**This study**
*H. boreorepandum*	H T. Niemela 1679	-	KX388658	-	Finland	[1]
*H. boreorepandum*	H 6003711	-	KX388657	-	Finland	[1]
*H. brevispinum*	IFP 019464	MW979559	MW980578	-	China	[6]
*H. brevispinum*	IFP 019465	MW979560	MW980579	-	China	[6]
*H. canadense*	H T N 09-006	-	KX388681	-	Canada	[1]
** *H. crassipedum* **	**FJAU66572**	**PV356811**	**PV329853**	**PP357260**	**China**	**This study**
** *H. crassipedum* **	**FJAU66573**	**PV356812**	**PV329854**	**PP357261**	**China**	**This study**
*H. cremeoalbum*	TUMH40462	-	AB906674	-	Japan	[1]
*H. cremeoalbum*	TUMH60740	-	AB906678	-	Japan	[1]
*H. cuspidatum*	RAS246	-	MH379944	-	USA	[2]
*H. cuspidatum*	RAS205	-	MH379936	-	USA	[2]
*H. ellipsosporum*	FD3281	KX086217	KX086215	-	Switzerland	[9]
*H. ellipsosporum*	H T. Niskanen 12-036	-	KX388671	-	Finland	[1]
*H. ferruginescens*	MH16005	-	MH379905	-	USA	[2]
*H. ferruginescens*	RAS229	-	MH379942	-	USA	[2]
*H. flabellatum*	IFP 019459	MW979556	MW980575	-	China	[6]
** *H. fulvostriatum* **	**FJAU66566**	**PV356807**	**PV329849**	**-**	**China**	**This study**
** *H. fulvostriatum* **	**FJAU66567**	**PV356808**	**PV329850**	**-**	**China**	**This study**
*H. ibericum*	BIO:Fungi:12330	-	HE611086	-	Spain	[8]
*H. ibericum*	MA-fungi 3457	-	AJ547879	-	Spain	[14]
*H. jussii*	IFP 019485	MW979539	MW980553	MW999436	China	[6]
*H. jussii*	IFP 019486	MW979540	MW980554	MW999437	China	[6]
*H. magnorufescens*	voucher 161209	KU612669	KU612549	-	Slovenia	[12]
*H. magnorufescens*	TO HG2818	-	KC293545	-	Italy	[15]
*H. melitosarx*	H T. Niskanen 11-056	-	KX388683	-	USA	[1]
*H. melitosarx*	K 176869	-	KX388685	-	UK	[1]
*H. melleopallidum*	SMI356	-	FJ845406	-	Canada	[15]
*H. mulsicolor*	LJU GIS 1336	-	AJ547885	-	Slovenia	[14]
*H. mulsicolor*	REB-341	-	JX093560	-	USA	[36]
*H. neorepandum*	H T. Niskanen 10-095	-	KX388659	-	Canada	[1]
*H. neorepandum*	H T. Niskanen 10-086	-	KX388660	-	Canada	[1]
*H. olympicum*	H T. Niskanen 09-134	-	KX388661	-	USA	[1]
*H. olympicum*	SAT-10-208-05	-	MT955159	-	USA	[6]
*H. oregonense*	HVM61	-	KF879509	-	USA	[37]
*H. oregonense*	PNW-MS g2010502h1-09	-	AJ534972	-	USA	[14]
** *H. orientalbidum* **	**FJAU66570**	**PV356809**	**PV329857**	**PP357258**	**China**	**This study**
** *H. orientalbidum* **	**FJAU66571**	**PV356810**	**PV329858**	**PP357259**	**China**	**This study**
*H. orientalbidum*	TUMH:64068	-	LC621862	-	Japan	[7]
*H. ovoideisporum*	voucher71106	-	KU612536	-	Slovenia	[6]
*H. ovoideisporum*	BIO Fungi 12683	-	NR119818	-	Spain	[8]
*H. pallidocroceum*	IFP 019466	MW979554	MW980568	MW999449	China	[6]
*H. pallidocroceum*	IFP 019467	MW979555	MW980569	MW999450	China	[6]
*H. pallidomarginatum*	IFP 019468	MW979552	MW980566	MW999447	China	[6]
*H. pallidomarginatum*	IFP 019469	MW979553	MW980567	MW999448	China	[6]
*H. pinicola*	SFC20180928-18	OR211401	OR211383	OR220059	Korea	[38]
*H. quebecense*	H T. Niskanen 10-064	-	KX388662	-	Canada	[1]
*H. quebecense*	CN9	-	MH379881	-	USA	[2]
*H. repando-orientale*	TUMH60745	-	AB906683	-	Japan	[39]
*H. repando-orientale*	TUMH60743	-	AB906684	-	Japan	[39]
*H. repandum*	H 6003710	-	NR164553	-	Finland	[1]
*H. roseotangerinum*	MHKMU LP Tang 3458	PQ287756	PQ287675	PQ295849	China	[22]
*H. roseotangerinum*	MHKMU LP Tang 3458-1	PQ287757	PQ287676	PQ295850	China	[22]
*H. rufescens*	H 6003708	-	KX388688	-	Finland	[1]
*H. rufescens*	HTN7839	-	KX388656	-	Estonia	[1]
*H. slovenicum*	LJU GIS 1338	-	AJ547870	-	Slovenia	[14]
*H. slovenicum*	LJU GIS 1340	-	AJ547884	-	Slovenia	[14]
*H.* sp.	wi1A4spel	-	KC679833	-	China	[6]
*H.* sp.	wi8T4spel	-	KC679834	-	China	[6]
*H.* sp.13	HKAS57714	KU612673	KU612617	-	China	[12]
*H.* sp.13	HKAS58838	KU612675	KU612616	-	China	[12]
*H.* sp.2	HKAS92340	KU612661	KU612543	-	China	[12]
*H. sphaericum*	IFP 019470	MW979549	MW980563	MW999444	China	[6]
*H. sphaericum*	IFP 019472	MW979550	MW980564	MW999445	China	[6]
*H. sphaericum*	IFP 019471	MW979551	MW980565	MW999446	China	[6]
*H. subberkeleyanum*	TNS:F-19323	-	LC621879	-	Japan	[7]
*H. subberkeleyanum*	TUMH 63627	-	LC621880	LC622505	Japan	[7]
*H. subconnatum*	RAS235	-	MH379930	-	USA	[2]
*H. subconnatum*	RAS169	-	MH379916	-	USA	[2]
*H. submulsicolor*	H T. Niskanen 10-132	-	KX388682	-	Canada	[1]
*H. subolympicum*	F1188765	KU612653	KU612599	-	USA	[12]
*H. subolympicum*	DAOM744368	-	MH174257	-	Canada	[1]
*H. subovoideisporum*	H 6003707	-	NR158494	-	Finland	[1]
*H. subrufescens*	H T. Niskanen 10-154	-	KX388649	-	Canada	[1]
*H. subrufescens*	F1188749	KU612663	KU612535	-	USA	[12]
*H. subtilior*	RAS180	-	MH379918	-	USA	[2]
*H. subtilior*	TENN 073034	-	NR164029	-	USA	[2]
*H. tenuistipitum*	IFP 019476	MW979557	MW980576	-	China	[6]
*H. tenuistipitum*	IFP 019477	MW979558	MW980577	-	China	[6]
*H. tomaense*	TUMH64085	-	LC622508	-	Japan	[7]
*H. tottoriense*	TUMH64088	-	LC621887	LC622511	Japan	[7]
*H. tottoriense*	TUMH64089	-	LC621888	LC622512	Japan	[7]
*H. treui*	KA20-0732	ON907772	ON907793	OR220061	Korea	[1]
*H. vagabundum*	10782TJB	-	MH379949	-	USA	[2]
*H. vagabundum*	CLO4985	-	MH379909	-	USA	[2]
*H. ventricosum*	IFP 019478	MW979547	MW980561	MW999442	China	[6]
*H. ventricosum*	IFP 019479	MW979548	MW980562	MW999443	China	[6]
*H. vesterholtii*	BIO:Fungi:10429	-	HE611084	-	Spain	[8]
*H. vesterholtii*	BIO:Fungi:10452	-	HE611085	-	Spain	[8]
*H. washingtonianum*	UBC F-32538	-	MF954990	-	Canada	[6]
*H. zongolicense*	GO-2010-142a	-	KC152121	-	Mexico	[1]
*Sistotrema. muscicola*	KHL 11721	-	AJ606040	-	Sweden	[40]
*Sistotrema. muscicola*	taxon:154757	-	AJ606041	-	Sweden	[40]

Newly generated sequences in this study are in bold.

## Data Availability

The species registration names and gene accession numbers referenced in this study are publicly available in the following online databases: Fungal Names (https://nmdc.cn/fungalnames/) and NCBI GenBank (https://www.ncbi.nlm.nih.gov/genbank/), with the data accessed on 25 March 2025.

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
