# Peer review of "New Contributions to the Species Diversity of the Genus Hydnum (Hydnaceae, Cantharellales) in China: Four New Taxa and Newly Recorded Species"

_jof, 2025, doi:10.3390/jof11060431_

Round 1

Reviewer 1 Report

I congratulate with the authors for the elegant revision of the genus Hydnum and the description of new species. 

No detailed comment

Author Response

Dear Reviewer,

Thank you for your careful review of our manuscript entitled "New Contributions to the Species Diversity of the Genus Hydnum (Hydnaceae, Cantharellales) in China: Four New Taxa and Newly Recorded Species" and for the valuable time you have dedicated to the evaluation process. In response to your suggestion regarding improving the English quality of the manuscript, we have carefully checked the entire text and made corresponding revisions. All modifications have been highlighted in either red or blue throughout the manuscript.

Please refer to the revised manuscript

Reviewer 2 Report

The authors have revealed the New Contributions to the Species Diversity of the Genus Hydnum (Hydnaceae, Cantharellales) in China: Four New Taxa and Newly Recorded Species, which submitted in the Journal of Fungi.

Introduction

Although it's pleasant and simple to follow, I didn't understand why the authors were interested in the genus.  There should be more explanation.

The methods for supporting genus identification for references 6-9, 12-24 are still absent, and the authors only displayed the number of species in China (Lines 47-49). Would you kindly include a detailed methodology? Do morphology and phylogenetic analysis employ them? For the reader, it is really pleasant.

Although these data in introduction are currently lacking, this review of significant research on morphology and phylogenetic approaches is crucial and simple to support the classification of species.

The gap analysis or research questions are not clear, and I suggested that the authors should be added in the last paragraph, possible before the objective.

Methods

Results and discussion

The detailed results are nice, they are interesting findings. However, I suggested that the comparative morphological data of Hydnum sp from Figs 1-7 should be added before Results 3.1 molecular phylogeny since the author mentioned in Line 21. All figures have no details and are unacceptable. The authors should included detailed anatomical labels for the mushrooms in accordance with the morphological text. Please ensure all images are with scale to make it clearer.

According to the morphological knowledge from this research, the concluded features (shape, size, average length, average width, and Q value) as the morphometric data and specimens examined data amongHydnum sp should be compared via the Table.

The authors mentioned the spore (Lines 65) or Basidiospores - Pileipellis were collected and why the scanning electric microscopy did not observed?

Such as the important previously published data

Khaund, P., & Joshi, S. R. (2014). Micromorphological characterization of wild edible mushroom spores using scanning electron microscopy. National Academy Science Letters37, 521-527.

Read, N. D., & Lord, K. M. (1991). Examination of living fungal spores by scanning electron microscopy.Experimental mycology15(2), 132-139.

Umroong, P. (2020). Techniques for Preparing Spores and Hyphae of Schizophyllum commune for Morphological Observation. Microscopy and Microanalysis Research–The Journal of The Microscopy Society of Thailand33(1), 22-27.

I read the discussion is based on poorly cited or previously reported findings. What are the new findings for the research and the sole potentially novel observation.

Conclusion

Could you please show the integrated data between morpho-phylogenetic analysis in conclusion? The authors should be highlighted and new findings.

Author Response

Dear Reviewers,

Thank you for your careful review and constructive comments on our manuscript titled "New Contributions to the Species Diversity of the Genus Hydnum (Hydnaceae, Cantharellales) in China: Four New Taxa and Newly Recorded Species". Your valuable suggestions have significantly improved the quality of our work. We have carefully addressed all comments and revised the manuscript accordingly. Below are our point-by-point responses:

Revision Notes: Revisions suggested by other reviewers in the manuscript have been highlighted in blue, while modifications made based on your valuable suggestions have been marked in red.

Reviewer 3 Report

Dear authors, I have provided comments for improvements on the manuscript. Please, note that two of them are a must:
1. Providing citations of relevant literature in the Introduction section;

2. Providing references for the published sequences in Table 1.

Please refer to the manuscript file for detailed comments.

Author Response

Dear Reviewers,

Thank you for your careful review and constructive comments on our manuscript titled "New Contributions to the Species Diversity of the Genus Hydnum (Hydnaceae, Cantharellales) in China: Four New Taxa and Newly Recorded Species". Your valuable suggestions have significantly improved the quality of our work. We have carefully addressed all comments and revised the manuscript accordingly. Below are our point-by-point responses:

Revision Notes: Revisions suggested by other reviewers in the manuscript have been highlighted in red, while modifications made based on your valuable suggestions have been marked in blue.

Round 2

Reviewer 2 Report

The author has addressed my comments and I have no questions.

The author has addressed my comments and I have no questions.